# Is There a Relationship between Antimicrobial Use and Antibiotic Resistance of the Most Common Mastitis Pathogens in Dairy Cows?

**DOI:** 10.3390/antibiotics12010003

**Published:** 2022-12-20

**Authors:** Zorana Kovačević, Marko Samardžija, Olga Horvat, Dragana Tomanić, Miodrag Radinović, Katarina Bijelić, Annamaria Galfi Vukomanović, Nebojša Kladar

**Affiliations:** 1Department of Veterinary Medicine, Faculty of Agriculture, University of Novi Sad, Trg Dositeja Obradovica 8, 21000 Novi Sad, Serbia; 2Clinic for Reproduction and Obstetrics, Veterinary Faculty, University of Zagreb, Heinzelova 55, 10000 Zagreb, Croatia; 3Department of Farmacology and Toxicology, Faculty of Medicine, University of Novi Sad, Hajduk Veljkova 3, 21000 Novi Sad, Serbia; 4Center for Medical and Pharmaceutical Investigations and Quality Control, Department of Pharmacy, Faculty of Medicine, University of Novi Sad, Hajduk Veljkova 3, 21000 Novi Sad, Serbia

**Keywords:** mastitis, cows, antimicrobial resistance, antimicrobial use

## Abstract

Antimicrobials have had an important impact on animal health and production performance. However, non-prudent antimicrobial use (AMU) in food producing animals is considered to contribute to the emergence of antimicrobial resistance (AMR), with a potential impact on both animal and public health. Considering the global importance of AMR, and the threats and challenges posed by mastitis and mastitis therapy in livestock production, the main objective of this study was to quantify AMU on three dairy farms in Serbia and to examine whether there is an association between AMU and the emergence of antimicrobial resistance of mastitis-associated pathogens. Antimicrobial susceptibility testing was performed by the disk diffusion method using causative agents isolated from the milk samples of 247 dairy cows. AMU data were obtained for a one-year period (May 2021 to May 2022) based on antibiotic prescriptions listed in electronic databases kept by farm veterinarians. To estimate antimicrobial drug exposure at the farm level, the veterinary drug Defined Daily Dose was calculated by multiplying the total amount of antibiotic used on the farms during the study period by the quantity of antibiotic in the administered drug and number of original drug packages used. The results on the association between the use of common antibiotics in mastitis treatment and AMR of isolated mastitis-associated pathogens confirm a pattern that could raise awareness of the importance of this aspect of good veterinary and clinical practice to combat the global threat of AMR.

## 1. Introduction

Antimicrobial resistance (AMR) affects both human and animal health and is one of the most important global issues in both human and veterinary medicine. It can spread from animals to humans, through the food chain or direct contact [1,2,3]. AMR is a public health threat due to the possible transmission of pathogens through food, and resistant strains of bacteria can also reduce the therapeutic effect of drugs used to treat animal and human infections [4,5]. Therefore, to address the challenges of AMR, a One Health approach is required, which places the focus on the connections between the human, animal, and environmental sectors [6]. As a vital component to national and global AMR strategies, the One Health strategy applies an interdisciplinary approach to surveillance and the implementation of programmes, policies, and research [7]. 

In order to combat AMR, the European Surveillance of Veterinary Antimicrobial Consumption (ESVAC) report was developed. This report provides insight into the data on antimicrobial use in animals from 31 European countries [8]. Though data from the Republic of Serbia, and many other developing countries, are not included, AMR is a particularly significant threat in these countries, not only because of the health issues they face, but also because of the increase in intensive small-scale livestock production that is exacerbated by poor sanitation infrastructure [9]. For these reasons, in 2019 Serbia adopted the National Antimicrobial Resistance Control Programme, which focuses special attention on monitoring the circulation and consumption of antimicrobials [10,11]. Moreover, in Serbia, the Medicines and Medical Devices Agency is responsible for the collection and processing of data on the use of drugs [12]. However, there is currently no information system or database for the follow-up of the drugs in veterinary medicine at the farm level. This makes it virtually impossible to trace the circulation and use of drugs, including antimicrobials, or to analyse the collected data and propose appropriate measures to prevent the spread of AMR. 

With an estimated 73% of the world’s total antibiotic use occurring in the livestock sector, the importance of reducing AMR is clear [13]. In intensive livestock production, mastitis is one of the greatest threats to health and welfare, causing huge economic losses in the dairy industry [14]. Etiological agents of this disease include various Gram-positive and gram-negative bacteria, some of which are infectious (e.g., *Staphylococcus aureus*, *Streptococcus agalactiae*, *Mycoplasma* spp.) while others are ubiquitous (e.g., *Escherichia coli*, *Enterococcus* spp., coagulase-negative *Staphylococcus*, *Streptococcus uberis*) [15]. Furthermore, improving biosecurity measures, milking hygiene, teat disinfection after milking, and the maintenance of milking machines should be part of control strategies to prevent mastitis [16,17]. However, antibiotic therapy (the local application of intramammary drugs and the systemic administration of antibiotics) is one of the main strategies to control mastitis [18]. Improper and irresponsible use of antibiotics, and the premature discontinuation of treatment, contribute to the development of resistant strains of bacteria [19], and the increasing number of clinically important pathogens developing resistance to antibiotics used in mastitis treatment is becoming critical [20]. According to some studies, there is a strong relationship between the use of antibiotics and the development and spread of AMR in the veterinary, agriculture, and medical sectors [21,22]. Therefore, it is important to investigate the connection between the use of antibiotics and AMR of the most important causative agent of mastitis. Based on the number of animals, it is only possible to determine the use of antimicrobial agents in milligrams of active substances per kilogram of animal weight (mg/PCU), without the possibility of a more precise analysis of the use of antibiotics in certain species and categories of animals [23]. 

The aim of present study was to investigate whether there is an association between AMU and AMR of the most common mastitis pathogens isolated from milk samples collected from dairy farms in the Republic of Serbia.

## 2. Results

### 2.1. Bacteriological Testing of Milk Samples

Figure 1 illustrates the prevalence of mastitis pathogens in milk samples from Farm 1. Based on the laboratory results, 33 of 66 (50%) milk samples were positive for mastitis-causing pathogens, of which 23 were clinical and 10 subclinical cases of mastitis, while the remaining 33 samples (50%) were negative. Of the isolated bacteriological causes of the mastitis, the most common pathogen was *Streptococcus* spp. identified in 11 samples (16.66%), *Escherichia coli* and *Serratia marcescens* in 8 samples (12.12%), *Staphylococcus aureus* in 3 samples (4.57%), and *Staphylococcus* spp. coagulase negative, *Streptococcus uberis* and *Klebsiella* spp. in 1 sample each (1.51%).

On Farm 2, bacteriological testing was performed on 99 milk samples. Pathogens were isolated in 65 samples (65.66%), of which 39 showed clinical and 26 subclinical mastitis. The most common pathogen was *Streptococcus* spp., identified in 19 samples (19.20%), followed by *E. coli* in 13 samples (13.13%), and β-haemolytic *Streptococcus* spp. in 8 samples (8.08%). Furthermore, *Staphylococcus* spp. coagulase negative was isolated in seven (7.07%), *S. uberis* in six (6.06%), and *S. aureus* in five (5.05%) samples. *Proteus mirabilis* (3.03%), *S. marcescens* (2.02%), *Staphylococcus* spp., and *K. oxytoca* (1.01%) were the least frequently identified causative agents (Figure 2).

A total of 82 milk samples were bacteriologically tested from Farm 3, and mastitis-causing pathogens were isolated from 46 (56.10%) samples, out of which 35 were clinical and 11 subclinical mastitis. *E. coli* was isolated in the majority of milk samples (26.82%). Twelve samples (14.70%) were identified as *Streptococcus* spp., followed by five samples (6.09%) of *S. aureus*, three samples (3.65%) of *S. uberis*. β-haemolytic *Streptococcus*, *Enterobacter sakazakii, S. marcescens* and *Streptococcus dysgalactiae* were present in only one sample each (1.21%) (Figure 3).

### 2.2. Antibiotics Use at the Farm Level

The data on antibiotics use to treat mastitis on all three farms and the calculated Defined Daily Dose (DDD) are shown in Table 1, Table 2 and Table 3. Cefalexin and cefquinome are used on all three farms as the most common antibiotic used in mastitis treatment, followed by enrofloxacin and amoxicillin+ clavulanic acid which are used on Farms 1 and 2. 

### 2.3. Multivariate Approach

Factor analysis of the dataset describing total antibiotics consumption (expressed in milligrams or IUs, depending on the drug) on the three farms indicates that after the extraction of principal components and varimax normalized rotation, the first two factor axes describe more than 98% of the dataset variability (Figure 4a). Most of the variability on the first axis (FA1) correlated with the recorded consumption of cloxacillin, penicillin, streptomycin, and marbofloxacin, and with the rate of application of cefalexin. On the other hand, the shape of the dataset variability was determined by the use of bacitracin, neomycin, tetracycline, and the combination of amoxicillin and clavulanic acid. The position of the evaluated farms in the space defined by the first two factor axes (Figure 4b) shows a grouping of Farm 1 and Farm 2 in the positive part of FA1 as a result of high use of antibiotics in the classes of cephalosporin, aminoglycosides, and fluoroquinolones, and the combination of amoxicillin and clavulanic acid. On the other hand, Farm 3 was positioned in the negative part of FA1 due to the more frequent application of penicillin, streptomycin, and combination of sulfamethoxazole and trimethoprim. However on the second axis (FA2), Farms 1 and 2 were separated as a consequence of the different choice of antibiotics in these classes, i.e., cefquinome and cefoperazone vs cefalexin, neomycin vs kanamycin.

Multivariate correspondent analysis of the data describing the susceptibility of bacterial isolates from the three farms (Appendix A) to evaluate the spectrum of antibiotics shows that the first two correspondent axes (dimensions) describe around 20% of the variability (inertia) (Figure 5). The position of Farm 2 in the positive part of the first correspondent axis is the result of domination of *Streptococcus* spp. (β-haemolytic *Streptococcus* spp., *S. dysgalactiae*, and *S. uberis*) and *Staphylococcus* spp. coagulase negative among the clinical isolates. Furthermore, these isolates show susceptibility to amoxicillin (AMX), ampicillin (AMP), cloxacillin (CLOXA), erythromycin (ERY), amoxicillin and clavulanic acid (AMC), ceftriaxone (CRO), and lincomycin (LCM), which correlates with the low frequency of application of these antibiotics (and other antibiotics belonging to the same class) on Farm 2 (Figure 4). On the other hand, Farms 1 and 3 were positioned closely in the negative part of the first correspondent axis. Clinical isolates from these farms are dominated by *Klebsiella* spp. (including *K. oxytoca*), *Staphylococcus* spp. (including *S. aureus*), *E. coli*, and *P. mirabilis*. These isolates were susceptible to treatment with tetracycline (TET), gentamycin (GEN), neomycin (NEO), trimethoprim/sulfamethoxazole (SXT), and streptomycin (STM), but on the other hand resistant to amoxicillin and clavulanic acid (AMC), amoxycillin (AMX), and cloxacillin (CLOXA), which are characterised by a high prescription rate on Farm 1 and Farm 3 (Figure 4).

## 3. Discussion

Antibiotic use in human medicine, veterinary medicine, and agriculture has been linked to the global rise of AMR. Moreover, interventions designed to reduce the use of antibiotics in food-producing animals have shown a positive effect in reducing the prevalence of AMR both in animals and humans in contact with those animals [39]. According to the joint scientific opinion of the European Medicines Agency (EMA) and the European Food Safety Authority (EFSA) on the use of antimicrobial agents in animal husbandry, it was noted that it is difficult to quantify the impact of a single factor due to the many factors contributing to AMR development, although there is evidence of a relationship between the reduction of AMU and the reduction of AMR [40]. Research focused on the correlations between veterinary AMU and AMR [39,41] have become more interesting, especially since it has been suggested that AMU in food animals can result in antibiotic-resistant infections in humans [42,43].

To date, no studies have focused on determining the relationship between AMR of the most common mastitis pathogens and AMU in mastitis treatment on dairy farms in the Republic of Serbia. Moreover, there is lack of such data in all countries of the Balkan region that have similar farm management practices as Serbia. Interestingly, with the aim of reducing bacterial resistance, guidelines and recommendations for the use of antibiotics in certain infections of different animal species have been drawn up in many countries to reduce irrational use that consequently leads to bacterial resistance [44]. However, such guidelines may have an issue with the origin of the information on bacterial resistance. Without systematic data collection, inconsistencies with local bacterial sensitivities can occur. Therefore, the basis of successful treatment of bacterial infections is sound knowledge of the local prevalence of causative agents and their degree of sensitivity, i.e., resistance. Regular monitoring of resistance levels and the structure of drug use, and comparison with use in other environments, can help to identify possible irregularities in the use of antibiotics. Due to the dynamic nature of bacteria, there is an increasing need to revise the guidelines and adapt them to local conditions. 

As such, this study is important as it gives insight into AMR patterns at the local farm level, while providing data on antimicrobial use in the same period. Additionally, antimicrobial susceptibility testing is an important component of prudent antimicrobial use practices [45]. For example, it has been shown that *Staphylococcus aureus* resistance for penicillin is higher on dairy farms that do not send milk samples for microbiological culture and susceptibility testing [46]. Interestingly, *Streptococcus* spp. and *E. coli* were the most dominant mastitis-causing pathogens on all three farms included in this study, which is in accordance with other studies in the Republic of Serbia [47,48], showing a high prevalence of resistance to many antibiotics, such as amoxycillin, ampicillin, and ceftriaxone. Moreover, continued use of antimicrobial susceptibility testing in research settings is recommended to monitor the trends in AMR among mastitis pathogens and to better understand potential factors relating to the persistence of infections and response to therapy [45]. Insight into the susceptibility of specific pathogens could help successfully guide the implementation of antimicrobial stewardship programmes.

The farms included in this study presented different levels of antimicrobial use as demonstrated by differences in DDD among farms and the amounts of antimicrobial drugs used to treat clinical or subclinical forms of mastitis. Pol and Ruegg [49] used a similar approach, where the use of DDD allows comparisons among compounds expressed in different units (i.e., mg and IU) and different administration routes, while also permitting the estimation of overall antimicrobial exposure at the farm level [49]. 

The multivariate analysis in our study showed certain patterns between frequently used antibiotics in mastitis treatment and AMR of isolated mastitis-associated pathogens on evaluated farms. Furthermore, *Streptococcus* spp. (including β-haemolytic *Streptococcus* spp., *S. dysgalactiae*, and *S. uberis*) and *Staphylococcus* spp. coagulase negative showed susceptibility to amoxicillin, ampicillin, cloxacillin, erythromycin, and combinations of amoxicillin and clavulanic acid, ceftriaxone, and lincomycin, which correlates with the low frequency of use of these antibiotics (or other antibiotics belonging to the same class) on Farm 2 (Figure 4). This susceptibility to antibiotics used in mastitis treatment is due to the fact that these are not frequently used in the treatment of mastitis in the Republic of Serbia [50]. On the other hand, a study in Brazil showed that although 94% of *S. uberis* clinical mastitis isolates were resistant to more than three antimicrobials, AMU was related to AMR only for tetracyclines, when the data were stratified according to antimicrobial classes [51]. Similar to veterinary medicine, monitoring the frequency of streptococcal resistance to antibiotics at the local level is essential for determining empirical therapy in human medicine. In a study on streptococcal isolates from inpatient and outpatient clinical samples in southeast Serbia, a very high resistance was found to macrolides in both inpatient and outpatient isolates of *S. pneumoniae*, *S. pyogenes*, and *S. agalactiae* (77.8%, 46.2%, and 32.4%, respectively), so these antibiotics should not be recommended for empirical therapy of infection caused by these bacteria. Moreover, all streptococci isolates were sensitive to vancomycin and linezolid, and all β -haemolytic streptococci isolates to penicillin and ceftriaxone [52]. The very high resistance to erythromycin among can be explained by uncontrolled and excessive consumption of total macrolides and long-acting macrolides (i.e., azithromycin) and other antibiotics in Serbia [53]. 

On the other hand, our study results showed that clinical isolates from Farm 1 and Farm 3 are dominated by *Klebsiella* spp. (including *K. oxytoca*), *Staphylococcus* spp. (including *S. aureus*), *E. coli*, and *P. mirabilis,* and are resistant to the combination of amoxicillin and clavulanic acid, and amoxicillin and cloxacillin, which are the most frequently used antibiotics on Farm 1 and Farm 3 (Figure 4). Contrary to our findings, the most common prescribed antibiotics in mastitis treatment in the Republic of Serbia are gentamicin, penicillin, streptomycin, cephalexin, sulfonamides, and enrofloxacin [50,54,55]. Interestingly, a study in Wisconsin hypothesised that the appropriate use of antimicrobial drugs should result in a reduction of the number of susceptible isolates within a population [49]. Moreover, in that same study, the relationships between the reported AMU and antimicrobial susceptibility in organic and conventional farms was determined since the use of two compounds commonly administered for treatment of IMI (penicillin and pirlimycin) was associated with the resistance of mastitis pathogens, though the use of many other commonly used compounds was not [49]. In Canada, a positive association was found in field conditions between AMR in bovine mastitis pathogens and herd-level AMU for mastitis treatment and control (intramammary penicillin and pirlimycin, as well as systemically administered penicillin and florfenicol) [56]. 

This study was conducted in accordance with the One Health strategy, which states that surveillance systems of monitoring infections need to be expanded to include antimicrobial use, as well as the emergence and spread of AMR within clinical and environmental samples [57]. Furthermore, according to the WHO’s call to apply surveillance to all sectors using the One Health approach, several countries in Europe now combine human, animal, food, and environmental data in their reporting, demonstrating how shared responsibility of health, agriculture, and environment authorities can result in robust surveillance systems [58]. Milk from cows with subclinical mastitis accidentally mixed into bulk milk enters the food chain and poses a threat to human health, since milk and milk products have the potential to transmit pathogenic organisms to humans [59].

In the future, establishing AMU monitoring programmes, prudent use guidelines, and educational campaigns are approaches that can minimise the further development of AMR [60]. In order to combat AMR, initiatives and significant efforts, such as the One Health initiative, are underway to curtail and optimise the use of critically important antimicrobials for human medicine in all applications, including food animal production [61]. Hence, special attention in the veterinary field should be focused on critically important antimicrobials for human medicine to minimise their use and consequently to protect public health. Concerns over the levels of AMU in agriculture and the potential association with the development and emergence of AMR will likely impact future drug use policy [45].

## 4. Materials and Methods

### 4.1. Sampling Procedures

The experimental protocol was approved by the Animal Ethics Committee of the Ministry of Agriculture, Forestry and Water Management, Veterinary Directorate (permit. no 9000–689/2, 6 July 2020). Milk samples were collected at three Holstein-Friesian dairy farms in Vojvodina province, Republic of Serbia, from May 2021 to May 2022. The number of cows on the farms ranged from 500 to 1500 cows. A total of 247 lactating cows were included in the study (Farm 1, *n* = 66; Farm 2, *n* = 99; Farm 3, *n* = 82) and selected for taking milk samples. The samples were taken from lactating animals with clinical and subclinical mastitis, 97 and 47, respectively. Cows were without other health problems, as confirmed by clinical examination. Clinical mastitis was diagnosed by examination of the udder by veterinarians on the farm, while subclinical mastitis was confirmed by the California Mastitis Test using somatic cell count in milk samples. The California mastitis test was carried out according to the method described by [62], at cowside by gently mixing an equal volume of milk with reagent (2 mL). Milk colour changes or formation of a viscose gel are readable within 1–2 min. Based on the reactions, the results were graded as negative (−), trace (T), weak positive (+), distinct positive (++), and strong positive reaction (+++). Samples for bacteriological testing were taken during morning milking. The milk samples were collected according to standard procedures, and prior to milking the udder skin was cleaned, washed, and dried and the first few streams of milk were discarded. Approximately 10 mL milk was collected and stored in sterile plastic tubes labelled with an ID number. All milk samples were stored in a cooler at 4 °C for transport to the Laboratory for Milk Hygiene, Department of Veterinary Medicine, Faculty of Agriculture, University of Novi Sad.

### 4.2. Isolation and Identification of Mastitis Pathogens and Antibiotic Susceptibility Pattern

Samples were incubated using a platinum loop (0.01 mL) for 48 h at 37 °C on 2% blood agar. Microorganisms growing on the plates were identified based on their biochemical and cultural traits. Isolation and identification of bacterial strains from collected milk samples were performed using microbiological procedures to diagnose mammary gland infection according to the National Mastitis Council [63]. A loopful milk sample was streaked on blood agar (Oxoid, Basingstoke, UK) and then subcultured on the following selective media: Mannitol salt agar, Edwards agar, Salmonella-Shigella agar, and MacConkey agar. The plates were incubated aerobically at 37 °C for 24 h. Plates were then examined for colony morphology, pigmentation, and haemolytic characteristics at 24–48 h. For distinguishing staphylococci and other Gram-positive cocci, the catalase test, mannitol fermentation test, coagulase test (either positive or negative), haemolytic pattern, and colony morphology were used. Isolates were confirmed by biochemical tests: oxidase activity, acid production (lactose sucrose and glucose fermentation), indole production, Voges–Proskauer, and hydrogen sulphide production. In addition, each strain was confirmed using Analytical Profile Index API-20 tests (API, bioMeraux, Craponne, France). To isolate staphylococci, the following media were used: blood agar, nutrient agar, Ziehl-Neelsen, and MSA. For *E. coli* isolation, nutrient agar, MacConkey agar, and API 25 were used. Regarding the phenotypic characteristics, the occurrence of α and β haemolysis was used for staphylococci, while pink colonies with precipitation were used for *E. coli*. Edwards agar and hydrolysis of esculin were used for streptococci determination.

Antimicrobial susceptibility testing was carried by in vitro disk diffusion (Kirby-Bauer) method on Mueller–Hinton agar (Oxoid) [64]. All isolated bacteria samples were tested using six commercially available antibiotic disks (Bioanalyse^®^, Ankara, Turkey) commonly used in veterinary practice, with the following disc potency: ampicillin (10 µg), streptomycin (10 µg), gentamicin (10 µg), trimethoprim/sulfamethoxazole (1.25/23.75 µg), enrofloxacin (5 µg), and ceftriaxone (30 µg). Each isolate was inoculated on nutrient broth and incubated aerobically overnight at 37 °C. After incubation, the bacterial suspension was vortexed and further diluted to a turbidity equivalent to that of 0.5 McFarland standards. The inoculum was spread on the surface of the Mueller–Hinton agar to achieve confluent bacterial growth. Antibiotic discs were immediately placed on the surface of the agar plate with sterile forceps, and incubated aerobically at 37 °C for 16 h. Resistance was determined by measurement of the inhibition of growth around the antimicrobial disk according to the zone diameter and interpreted as sensitive, intermediate, or resistant according to the Clinical Laboratory Standards Institute [65,66].

### 4.3. Data on Antibiotics Use at the Farm Level and Calculation of the Defined Daily Dose

Collection of data on antibiotic use was carried out on the same three farms where milk samples were collected over the same one-year period (May 2021 to May 2022). All prescriptions were written in an electronic database along with the clinical history of animals and diagnostic work-up, allowing for a computer search and transfer of information to spreadsheets (Excel version 9.0). A computer search identified all animals diagnosed with mastitis and treated with antibiotics during the study period. The following information were gathered: drug brand name, active substance, pharmaceutical form, dosage, duration of treatment, and route of administration. To estimate antimicrobial drug exposure at the farm level, the Defined Daily Dose for each veterinary drug was characterised as the maximum dose that a standard animal (BW = 680 kg) would receive if it was treated following the label dosages of the Summary of the Product Characteristics of the antimicrobial drug approved by the Agency of Medicine and Medical Devices of Republic of Serbia. Defined Daily Doses (DDD) were calculated using the following formula: DDD_A_ = MG_DDDA_ × U_DDDA_ × F_DDDA_
where DDD_A_ is the DDD for the antimicrobial “A”, MG_DDDA_ is the dose (mg or IU) contained in 1 mL or in an intramammary syringe of compound “A”, U_DDDA_ is the number of millilitres used in each administration, and F_DDDA_ is the frequency, or number of times per day that the compound is administered [49].

Further, the total amount of antibiotics used on the farms during the study period was calculated by multiplying the amount of antibiotic in the administered drug and number of original drug packages used. This was further processed to create a database containing the internationally non-proprietary names (INN) of antibiotics used on all three farms, and the quantities of pharmacological active substances (expressed in milligrams or IUs) administered at each farm (data are calculated from Table 1, Table 2 and Table 3, Appendix A) during the study period.

### 4.4. Data Analysis

The results obtained were summarized by Microsoft Office Excel (v2019) and statistically analysed by Tibco Statistica (v13.5). Depending on the variable type, different statistics approaches were used. Descriptive statistics were applied to determine the incidence of pathogens isolated on the farms. To better understand the patterns of antibiotics usage and recorded AMR to antibiotics on the evaluated farms, multivariate statistical methods (factor analysis and multiple correspondence analysis) were used. Both applied multivariate techniques are dimension reduction techniques that enable a better understanding of dataset patterns of variability in the space described by a lower number of dimensions. In factor analysis, the newly calculated dimensions are called factors, while in the case of correspondent analysis they are called correspondent axes. The calculated dimensions are correlated with original variables used to describe the specific dataset.

Factor analysis is applied to continuous variables and in this study was applied to the dataset describing the amounts of antibiotics used at the farms to depict the grouping of specific farms and the dominant antibiotics prescribed. On the other hand, multivariate correspondent analysis is applied to categorical variables and here was applied to the dataset describing resistance (coded as susceptible, intermediate, and resistant) of clinical bacterial isolates on the three farms, thus showing associations of recorded AMR with specific farms.

### 4.5. Limitations of the Study

Interpretation of the findings of this study should take into account certain potential limitations that might affect its conclusions. Additional guidelines for mastitis treatment are needed to curtail the use of antibiotics, and such guidelines have helped to reduce the use of antimicrobial use in food animal production. Currently, groups of researchers from all over Europe are working on six different guidelines on AMU [67]. One will be developed for mastitis treatment and will provide stricter regulations on AMU in animal farming, which will hopefully be an effective measure to reduce AMU. Moreover, additional research is needed on enhancing biosecurity to prevent disease, coupled with more judicious use of antibiotics used to treat and control infectious diseases in food animals. 

## 5. Conclusions

This study confirms the existence of patterns between frequently used antibiotics in mastitis treatment and AMR of isolated mastitis-associated pathogens on the evaluated farms. These patterns make this study interesting for future strategies aimed at the control of mastitis, although more data are needed to better understand how antimicrobials are used in mastitis to find opportunities for further reduction.

## Figures and Tables

**Figure 1 antibiotics-12-00003-f001:**
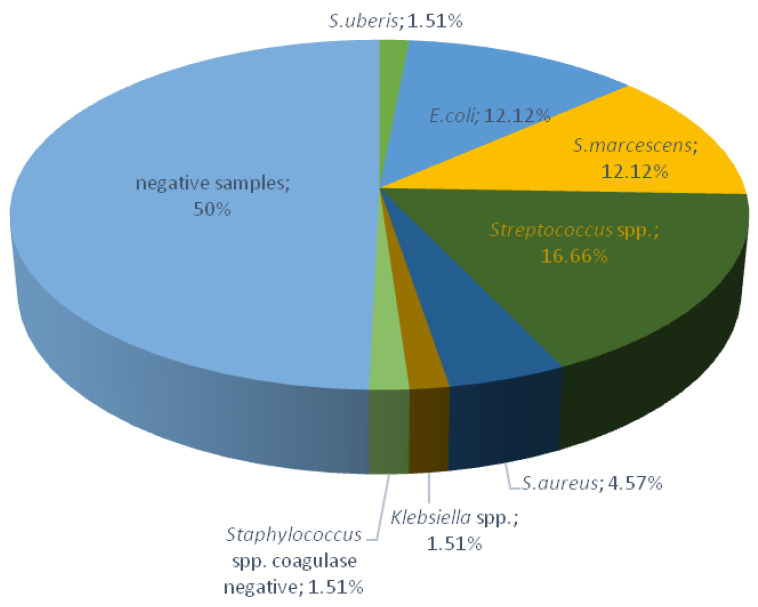
Prevalence of mastitis-causing pathogens in collected milk samples on Farm 1.

**Figure 2 antibiotics-12-00003-f002:**
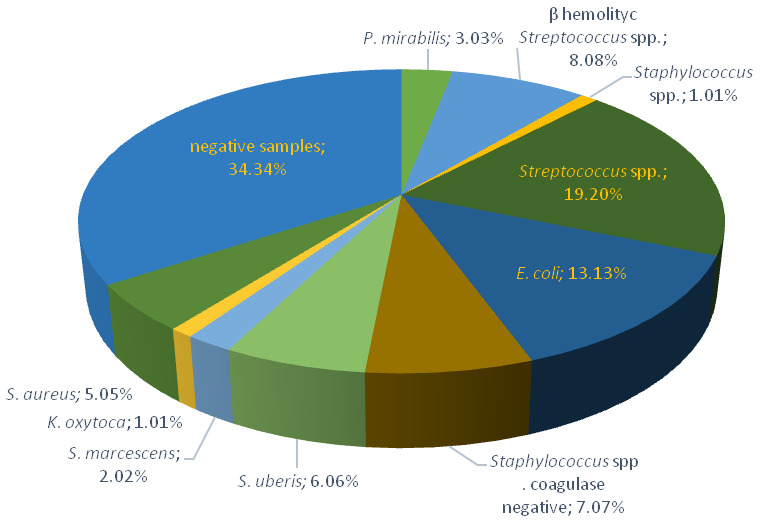
Prevalence of mastitis-causing pathogens in collected milk samples on Farm 2.

**Figure 3 antibiotics-12-00003-f003:**
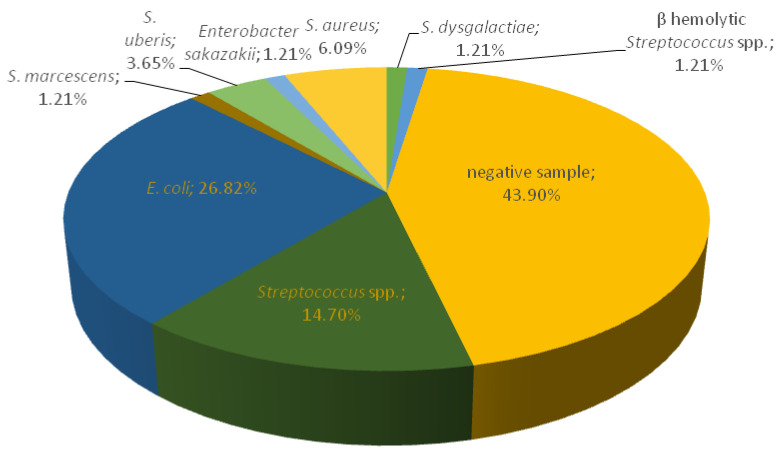
Prevalence of mastitis-causing pathogens in collected milk samples on Farm 3.

**Figure 4 antibiotics-12-00003-f004:**
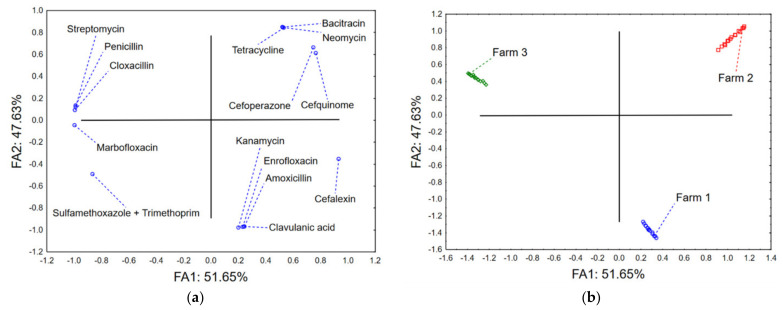
(**a**) Loadings of the first two factor axes and (**b**) position of the farms in the space defined by the first two factor axes.

**Figure 5 antibiotics-12-00003-f005:**
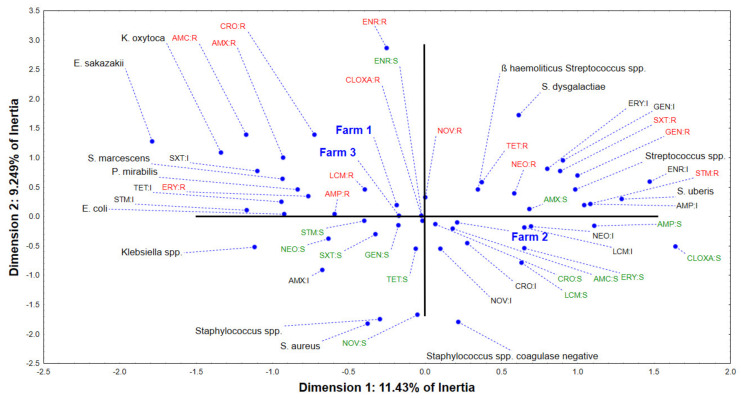
The position of the evaluated variables in the space defined by the first two correspondent axes.

**Table 1 antibiotics-12-00003-t001:** Defined Daily Doses (DDD) and quantity of antimicrobial drugs on Farm 1.

Brand Name	Intramammary Antibiotic	Amount per Syringe or mL	Number of Syringe or mL	Dose *	Frequency, Times/Day	DDD	Quantity
	Intramammary dry cow therapy						
Drycloxa-kel^®^ [24]	cloxacillin	1000 mg	4	-	1	4000 mg	36 im
Rilexine 500^®^ [25]	cefalexin	375 mg	1	-	1	375 mg	11 im
	Intramammary lactating mastitis						
Cefimam^®^ [26]	cefquinome	75 mg	3	-	1	225 mg	192 im
Dairymed^®^ [27]	amoxicillin	200 mg	3	-	1	600 mg	5 im
	clavulanic acid	50 mg	3	-	1	150 mg	
Mastijet^®^ forte [28]	tetracycline	200 mg	2	-	2	800 mg	20 im
	neomycin	250 mg	2	-	2	1000 mg	
	bacitracin	2000 IU	2	-	2	8000 IU	
Rilexine^®^ 200 [29]	cefalexin	200 mg	2	-	2	800 mg	195 im
Ubrolexin^®^ [30]	cefalexin	200 mg	2	-	1	400 mg	40 im
	kanamycin	100,000 IU	2	-	1	200,000 IU	
	Parenteral therapy						
Cobactan^®^ [31]	cefquinome	25 mg	-	1 mg/kg	1	680 mg	37 inj
Enrocin S^®^ [32]	enrofloxacin	100 mg	-	5 mg/kg	1	3400 mg	24 inj
Kelbomar^®^ [33]	marbofloxacin	100 mg	-	2 mg/kg	1	1360 mg	36 inj
Synulox^®^ RTU [34]	amoxicillin	140 mg	-	7 mg	1	4760 mg	39 inj
	clavulanic acid	35 mg	-	1.75 mg	1	1190 mg	

* Cow weight was assumed to be 680 kg; im—intramammary injector; inj—solutio for injectiones.

**Table 2 antibiotics-12-00003-t002:** Defined Daily Doses (DDD) and quantity of antimicrobial drugs on Farm 2.

Brand Name	Intramammary Antibiotic	Amount per Syringe or mL	Number of Syringe or mL	Dose * (mg/kg)	Frequency, Times/Day	DDD	Quantity
	Intramammary dry cow therapy						
Rilexine 500^®^ [25]	cefalexin	375 mg	1	-	1	375 mg	20 im
	Intramammary lactating mastitis						
Mastijet forte^®^ [28]	tetracycline	200 mg	2	-	2	800 mg	1588 inj
	neomycin	250 mg	2	-	2	1000 mg	
	bacitracin	2000 IU	2	-	2	8000 IU	
Dairymed^®^ [27]	amoxicillin	200 mg	3	-	1	600 mg	54 inj
	clavulanic acid	50 mg	3	-	1	150 mg	
Pathozone^®^ [35]	cefoperazone	250 mg	1	-	1	250 mg	6 inj
	Parenteral therapy						
Cobactan^®^ [31]	cefquinome	25 mg	-	1	1	680 mg	7920 mL
Rilexine^®^ [36]	cefalexin	150 mg	-	15	1	10,200 mg	1200 mL
Baytril^®^ MAX [37]	enrofloxacin	100 mg	-	5	1	3400 mg	90 mL

* Cow weight was assumed to be 680 kg; im—intramammary injector; inj—solutio for injectiones.

**Table 3 antibiotics-12-00003-t003:** Defined Daily Doses (DDD) and quantity of antimicrobial drugs on Farm 3.

Brand Name	Intramammary Antibiotic	Amount per Syringe or mL	Number of Syringe or mL	Dose * (mg/kg)	Frequency, Times/Day	DDD	Quantity
	Intramammary dry cow therapy						
Drycloxa-kel^®^ [24]	cloxacillin	1000 mg	4	-	1	4000 mg	156 im
	Intramammary lactating mastitis						
Cefimam^®^ [26]	cefquinome	75 mg	3	-	1	225 mg	1296 im
Mastijet forte^®^ [28]	tetracycline	200 mg	2	-	2	800 mg	37 im
	neomycin	250 mg	2	-	2	1000 mg	
	bacitracin	2000 IU	2	-	2	8000 IU	
	Parenteral therapy						
Kelbomar^®^ [33]	marbofloxacin	100 mg	-	2	1	1360 mg	93 inj
Penstrep^®^ [38]	penicillin	200 mg	-	8	1	5400 mg	298
	streptomycin	250 mg	-	10	1	6800 mg	

* Cow weight was assumed to be 680 kg; im—intramammary injector; inj—solutio for injectiones.

## Data Availability

The data used to support the findings of this study are available in the present manuscript.

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
