# Peer review of "Is There a Relationship between Antimicrobial Use and Antibiotic Resistance of the Most Common Mastitis Pathogens in Dairy Cows?"

_antibiotics, 2022, doi:10.3390/antibiotics12010003_

Round 1

Reviewer 1 Report

in the abstract on lines 25-27 the authors could put the name of the methodology used
in line 40 the authors can introduce the "one heath" concept or how antibiotic resistance impacts public/human health
on line 73 the authors can place a bibliographical reference to support the proposed strategies

the authors should associate the findings with the one health concept or the impact on public/human health
 may also associate/discuss the findings with the emergence of resistant streptococci in hospitals, for example

Author Response

Thank you for your insightful comments and suggestions that helped us significantly improve the quality of our manuscript. We strongly believe we have managed to address each of your concerns and comments. We have addressed each of concerns as outlined below and we tried to state point-by-point the changes we have made to the manuscript. Regarding better understanding, all changes in the text are marked in red color. Furthermore, the manuscript is reviewed by a native English speaker and it is given in track changes.

Point 1: In the abstract on lines 25-27 the authors could put the name of the methodology used

Response 1: The sentences in Lines 25-31 are changed and added according to your suggestion explaining the methodology used in the manuscript.

Point 2: In line 40 the authors can introduce the "one heath" concept or how antibiotic resistance impacts public/human health

Response 2:  Regarding your useful comment, "one heath" concept is introduced in Lines 46-51.

Point 3: On line 73 the authors can place a bibliographical reference to support the proposed strategies

Response 3: Regarding your comment, we added bibliographical references to support the proposed strategies in Line 79.

Point 4: The authors should associate the findings with the one health concept or the impact on public/human health

Response 4: Thank you for noticing this. We associated our findings in the following part of the discussion in Lines 320-331.

Point 5: May also associate/discuss the findings with the emergence of resistant streptococci in hospitals, for example

Response 5: As you suggested, we have associated the resistance of streptococci and AMU in inpatients and outpatients in Serbia with our findings related to streptococci resistance and AMU on Farm discussion section in Lines 288-300.

Reviewer 2 Report

The manuscript addresses an important issue related to antibiotic resistance of mastitis pathogens in three dairy farms in Serbia. Although the study is important and could be valuable for the effective management and treatment of mastitis in dairy cows, I have several major concerns related to the lack of methodology in this study. Please find my comments below:

The introduction should be rewritten as the authors repeated the same text or idea again and again which makes it boring for the reader.

My major concerns were about the identification of pathogens. The authors stated that the milk samples were inoculated on 2% blood agar to identify different types of bacteria in the milk samples. Unfortunately, blood agar can identify certain bacteria but not all the bacterial strains reported in the present study. It was also mentioned that biochemical and cultural characteristics were considered, but no details were given. To increase the reader's confidence in your findings, please revise the text and provide detailed information on how the bacteria were characterised. Also, include microscopic images to identify the different bacterial strains.

Another important point: The way the cows were classified in CM and SCM is not convincing. The statement was general, and I could not find out if the California Mastitis Test or SCC was used to classify the health status of the udder!!!! This was reflected in the mastitis identification results where the authors found that almost 50% of the samples they collected as mastitis milk were healthy and negative for bacteria.

If some of the samples you selected as mastitis milk are negative for bacteria, it means that there was a problem with the method you used to classify the animals as healthy, SCM and CM. The authors may not be aware that there are several factors that affect the number of SCC in the milk, such as season, parity, milk production, diurnal variations, etc. Therefore, not every increase in SCC indicates SCM. Therefore, this point should be discussed to justify the reported results.

Furthermore, the authors claim that "the samples were taken from lactating animals with clinical and subclinical mastitis without other health problems". How were other health problems excluded? Please provide information on the criteria used to confirm that the animals had no other health problems.

The statistical analysis is not clear. Please provide more information.

Author Response

Thank you for your insightful comments and suggestions that helped us significantly improve the quality of our manuscript. We believe we have managed to address each of your concerns and comments. We have addressed each of concerns as outlined below and we tried to state point-by-point the changes we have made to the manuscript. Regarding better understanding, all changes in the text are marked in blue color, except Point 6 that is given in green color. Furthermore, the manuscript is reviewed by a native English speaker and it is given in track changes.

Point 1: The introduction should be rewritten as the authors repeated the same text or idea again and again which makes it boring for the reader.

Response 1: Regarding your useful comment, we have rewritten the introduction.

Point 2: My major concerns were about the identification of pathogens. The authors stated that the milk samples were inoculated on 2% blood agar to identify different types of bacteria in the milk samples. Unfortunately, blood agar can identify certain bacteria but not all the bacterial strains reported in the present study. It was also mentioned that biochemical and cultural characteristics were considered, but no details were given. To increase the reader's confidence in your findings, please revise the text and provide detailed information on how the bacteria were characterised. Also, include microscopic images to identify the different bacterial strains.

Response 2:  In accordance with your useful suggestion, the text was added in Lines 372-394. Actually, methodology used in this paper is also used in our previously published papers (Kovačević, Z.; Radinović, M.; Čabarkapa, I.; Kladar, N.; Božin, B. Natural agents against bovine mastitis pathogens. Antibiotics 2021, 10, 205.;

Kovačević, Z.; Kladar, N.; Čabarkapa, I.; Radinović, M.; Maletić, M.; Erdeljan, M.; Božin, B. New perspective of Origanum vulgare L. and Satureja montana L. essential oils as bovine mastitis treatment alternatives. Antibiotics 2021, 10, 1460.).

Furthermore, regarding the microscopic images of the different bacterial strains, they are not included into the standard procedure of their identification, so they are not part of the this manuscript.

Point 3: The way the cows were classified in CM and SCM is not convincing. The statement was general, and I could not find out if the California Mastitis Test or SCC was used to classify the health status of the udder!!!! This was reflected in the mastitis identification results where the authors found that almost 50% of the samples they collected as mastitis milk were healthy and negative for bacteria.

Response 3: Regarding your comment we added explanation in Lines 352-361 for performing California Mastitis Test in our study. Furthermore, mastitis milk do not have positive bacteriological finding in 100% cases, so samples taken from cows with mastitis can be negative as found by other author also (Castañeda Vázquez, H., Jäger, S., Wolter, W., Zschöck, M., Vazquez, C., & El-Sayed, A. (2013). Isolation and identification of main mastitis pathogens in Mexico. Arquivo Brasileiro de Medicina Veterinaria e Zootecnia, 65, 377-382.).

Point 4: If some of the samples you selected as mastitis milk are negative for bacteria, it means that there was a problem with the method you used to classify the animals as healthy, SCM and CM. The authors may not be aware that there are several factors that affect the number of SCC in the milk, such as season, parity, milk production, diurnal variations, etc. Therefore, not every increase in SCC indicates SCM. Therefore, this point should be discussed to justify the reported results.

Response 4:   Factors affecting SCC, that you have mentioned are very important and can lead to increase of SCC but in lesser amount than intramammary infection what is described by Harmon 1994 (Harmon, R. J. (1994). Physiology of mastitis and factors affecting somatic cell counts. Journal of dairy science, 77(7), 2103-2112. Journal of dairy science vol 77, issue 7, 1994.). Furthermore, form of mastitis (clinical or subclinical) is added in Supplementary material (Supplementary 1) for each cow included in the study.

Point 5: Furthermore, the authors claim that "the samples were taken from lactating animals with clinical and subclinical mastitis without other health problems". How were other health problems excluded? Please provide information on the criteria used to confirm that the animals had no other health problems.

Response 5: Regarding your suggestion, we added explanation. Actually, cows used in the study were without other health problems what was confirmed with clinical examination. It is added in Lines 353-354.

Point 6: The statistical analysis is not clear. Please provide more information.

Response 6: According to your useful comment, we made correction. More details are now presented in Material and Methods section in Lines 446-466.

Reviewer 3 Report

In the manuscript antibiotics-2086583, authors aimed to show is there interconnection and to what extend between AMU on the different farms and AMR of the most common mastitis pathogens isolated from milk samples collected from those dairy farms located in the Republic of Serbia.

Generally, the paper topic is moderate interesting as the study try to confirm existence of the patterns between frequently used antibiotics in mastitis treatment and AMR of isolated mastitis associated pathogens on evaluated farms.

I suggest a major revision because the current manuscript lacks several qualities that are considered adequate for publication, as follows:

  1. The manuscript is technically sound, and the data do support the conclusions. "I suggest authors add a new subtitle before the conclusion, "Limitations of the Study," and elaborate on the current limitation or limitations," lines 380–382.
  2. The statistical analysis has been performed appropriately, but it is unclear.

·        - Lines 371-372: Data were evaluated by application of descriptive and multivariate statistical analysis; how did the authors generalize the obtained data? Lines 300–301 make it clear that the number of lactating cows used on each farm is not equal. Before the authors entered the statistical analysis, I assume the used data was still not in a normal distribution.

·        - In line 303, the authors did not specify how many clinical or subclinical mastitis were used from each farm.

  1. The manuscript is presented in a less than intelligible fashion and written in unclear and ambiguous standard English, especially in the introduction and discussion. I suggest the authors revise the manuscript with a concise introduction and related discussion (if this study was only recently conducted in Serbia, how about other countries? Try to discuss somewhat similar countries that have similar farm management practices for using antibiotics. In addition, I recommend that your manuscript be reviewed by a native English speaker before submission.
  2. Additional comments: check again the supplementary materials, is it necessary to split into two files? Give the note for reading S, I, and R in Supplementary 1. 

Author Response

Thank you for your significant and valuable opinion regarding our manuscript. Regarding better understanding, all changes in the text are marked in green color.

Point 1: The manuscript is technically sound, and the data do support the conclusions. "I suggest authors add a new subtitle before the conclusion, "Limitations of the Study," and elaborate on the current limitation or limitations," lines 380–382.

Response 1: Regarding your comment, new subtitle is added before the conclusion, "Limitations of the Study,". Part of the conclusion is transferred from the conclusion to this section, while new text is added in Lines 468-469 and 472-476.

Point 2: The statistical analysis has been performed appropriately, but it is unclear.

Response 2: According to your useful comment, we made correction. More details are now presented in Material and Methods section in Lines 446-466.

Point 3: Lines 371-372: Data were evaluated by application of descriptive and multivariate statistical analysis; how did the authors generalize the obtained data? Lines 300–301 make it clear that the number of lactating cows used on each farm is not equal. Before the authors entered the statistical analysis, I assume the used data was still not in a normal distribution.

Response 3: According to your useful comment, we made correction. Specifically, descriptive statistics was used for reporting the incidence of clinically isolated pathogens on evaluated farms. On the other hand, the multivariate correspondent analysis is applied on categorical data, meaning that the normality of distribution is irrelevant.

Point 4: In line 303, the authors did not specify how many clinical or subclinical mastitis were used from each farm.

Response 4: Regarding your comment, we added in Lines 352-353 as well as in Results section, Lines 106-107; 121-122 and 135-136 number of CM and SCM cases. Furthermore, form of mastitis (clinical or subclinical) is added in Supplementary material (Supplementary 1) for each cow included in the study.

Point 5: The manuscript is presented in a less than intelligible fashion and written in unclear and ambiguous standard English, especially in the introduction and discussion. I suggest the authors revise the manuscript with a concise introduction and related discussion (if this study was only recently conducted in Serbia, how about other countries? Try to discuss somewhat similar countries that have similar farm management practices for using antibiotics. In addition, I recommend that your manuscript be reviewed by a native English speaker before submission.

Response 5: Regarding your comment, we added explanation in Lines 233-235 that there is lack of these data in all countries on Balkan region as countries that have similar farm management practices as Serbia. Moreover, we compare our results in discussion with other countries, such as Brazil (in Lines 285-288) and Canada (in Lines 315-319).

According to your suggestion, manuscript is reviewed by a native English speaker and it is given in track changes.

Point 6: Check again the supplementary materials, is it necessary to split into two files? Give the note for reading S, I, and R in Supplementary 1.

Response 6: Thank you for your useful comment. Our opinion is that the supplementary materials have to be split into two files regarding the better understanding. Furthermore, the note for reading S, I, and R in Supplementary 1 is given (S for sensitive, I for intermediate and R for resistant).

Round 2

Reviewer 1 Report

the authors answered the questions and made the changes in the manuscript.

Reviewer 2 Report

The authors failed to justify the inquiries of the reviewer. There are significant problems in the methodology of this paper including the method used to identify the health status of the udder and the subsequent selection of the samples. Also, their described methods for identifying various pathogens are not trustworthy. 

Reviewer 3 Report

Dear Authors

I have taken note of the effort you have put into this revision; all comments and concerns have been addressed and justified; therefore, it could be accepted for publication.

Best regards